# Transmission matrix parameter estimation of COVID-19 evolution with age compartments using ensemble-based data assimilation

Santiago Rosa[1,2]*, Manuel A. Pulido[2,3], Juan J. Ruiz[4,5], Tadeo J. Cocucci[1,2]

**1** FaMAF, Universidad Nacional de Córdoba, Córdoba, Córdoba, Argentina, **2** FaCENA, Universidad Nacional del Nordeste, Corrientes, Corrientes, Argentina, **3** CNRS - IRD - CONICET - UBA, Instituto Franco-Argentino para el Estudio del Clima y sus Impactos (IRL 3351 IFAECI), Buenos Aires, Argentina, **4** CONICET - Universidad de Buenos Aires, Centro de Investigaciones del Mar y la Atmósfera (CIMA), Buenos Aires, Argentina, **5** Departamento de Ciencias de la Atmósfera y los Océanos, FCEN, Universidad de Buenos Aires, Ciudad Autónoma de Buenos Aires, Buenos Aires, Argentina

* santiago.rosa@mi.unc.edu.ar

**Data availability statement:** The raw data used in this work can be found at https://sisa.msal.gov.ar/datos/descargas/covid-19/files/Covid19Casos.zip. A curated

## Abstract

The COVID-19 pandemic, with its multiple outbreaks, has posed significant challenges for governments worldwide. Much of the epidemiological modeling relied on pre-pandemic contact information of the population to model the virus transmission between population age groups. However, said interactions underwent drastic changes due to governmental health measures, referred to as non-pharmaceutical interventions. These interventions, from social distancing to complete lockdowns, aimed to reduce transmission of the virus. This work proposes taking into account the impact of non-pharmaceutical measures upon social interactions among different age groups by estimating the time dependence of these interactions in real time based on epidemiological data. This is achieved by using a time-dependent transmission matrix of the disease between different population age groups. This transmission matrix is estimated using an ensemble-based data assimilation system applied to a meta-population model and time series data of age-dependent accumulated cases and deaths. We conducted a set of idealized twin experiments to explore the performance of different ways in which social interactions can be parametrized through the transmission matrix of the meta-population model. These experiments show that, in an age-compartmental model, all the independent parameters of the transmission matrix cannot be unequivocally estimated, i.e., they are not all identifiable. Nevertheless, the time-dependent transmission matrix can be estimated under certain parameterizations. These estimated parameters lead to an increase in forecast accuracy within age-group compartments compared to a single-compartmental model assimilating observations of age-dependent accumulated cases and deaths in Argentina. Furthermore, they give reliable estimations of the effective reproduction number. The age-dependent data assimilation and forecasting of virus transmission are crucial for an accurate prediction and diagnosis of healthcare demand.

version is available at
http://covid19.unne.edu.ar/obs_arg.csv. The
codes developed for the work are available at
https://gitlab.com/pulidom/covid/.

**Funding:** This study received funding from the
Ministry of Science, Technology and Innovation
of Argentina (PICT2021-I130 Dr. Manuel A.
Pulido, Dr. Juan J. Ruiz), the General Secretariat
for Science and Technology, Universidad
Nacional del Nordeste (CORR 01 COVID
FEDERAL EX-2020-38902538, Santiago Rosa,
Dr. Manuel A. Pulido, Dr. Juan J. Ruiz), and the
National Scientific and Technical Research
Council (PICT2020-SERIEA-I-A Dr. Juan J.
Ruiz). The sole responsibility for the content of
this publication lies with the authors. The
funders had no role in study design and
analysis, decision to publish, or preparation of
the manuscript.

**Competing interests:** The authors have
declared that no competing interests exist.

# 1 Introduction

Governments worldwide faced several challenging decisions as the SARS-COV-2 virus spread in early 2020. Several non-pharmaceutical interventions, from social distancing measures for high-risk population to general lockdowns, were implemented to alleviate the propagation of COVID-19, at the expense of a decline in productivity. While lockdowns can significantly impact epidemic propagation by flattening the active cases curve, they also negatively affect education and social activities. Moreover, COVID-19 outbreaks impact the economy, as evidenced in the case of strictly enforced sick leaves. Therefore, decision-makers must carefully evaluate the trade-off between socio-economical well-being and public health. Real-time decision-making tools are required for monitoring the pandemic's situation and for predicting the evolution of the disease at different scales: from neighborhoods and cities to states and nationwide. Epidemiological predictions can help prevent the health system overload, allowing governments to implement timely non-pharmaceutical interventions and avoid healthcare collapse. Research on monitoring and modeling COVID-19 spread (e.g.[1]) had a strong political impact worldwide. However, the dispair COVID-19 evolution in various countries made clear that continuous monitoring of the local spreads was required to adopt timely distancing measures.

The propagation of COVID-19 has been modeled using epidemiological compartmental models, such as Susceptible-Exposed-Infected-Recovered (SEIR) models. The exponential growth of the initial phase of an outbreak may be well represented by compartmental models. However, the virus propagation is subject to the complexity of human interactions or individual-wise varying viral loads [2] and this poses challenges for compartmental models to accurately describe such dynamics. Even the most advanced meta-population models (e.g. GLEAM [3]) and agent-based models [4] crudely represent the transmission dynamics of the virus due to the inherent difficulty in modeling interactions between individuals. Furthermore, social life, and so human interactions, underwent significant changes throughout the pandemic.

The accumulated data on the epidemic was rather limited and prone to errors due to detection policies changed with time, delays in reported cases occurring during weekends, and the absence of hospital discharge dates, among other factors. In addition to these sources of data uncertainty, a significant number of cases were not detected. Many individuals either experienced mild or no noticeable symptoms so that they were not reported and, on a smaller scale, the tests gave false negatives [5]. Given the incomplete and noisy nature of the data and the inherent challenges in accurately representing complex underlying processes with models, the idea of combining model and data becomes appealing. Real-time model-data fusion techniques, such as sequential inference and data assimilation, aim to combine very diverse sources of information considering their uncertainties.

There are several ways to combine data with models during the evolution of an outbreak. In the context of Bayesian inference, some works use Monte Carlo Markov-Chain models [6–9]. Alternatively, other works propose the use of data assimilation techniques for epidemiological modeling, which is computationally cheaper at the cost of assuming Gaussianity. Shaman et al. [10,11], use an ensemble-based data assimilation framework to model influenza propagation. The state evolution of an epidemiological SIRS model (Susceptible, Infectious, Recovered, Susceptible), is combined with direct and indirect data (e.g. level of web activity related to the illness) from the epidemic. At the same time, the parameters of the system are learned online as the observations become available. In these works, they use a variant of the ensemble Kalman filter (EnKF). Due to the need for monitoring the spread of COVID-19

and the abundance of worldwide data, some works use these data assimilation techniques to estimate the spread of the SARS-COV-2 virus. Li et al. [12] use the iterated filter-ensemble adjustment Kalman filter to assimilate COVID-19 data within China using a meta-population model and mobility data. They propose the estimation of the undocumented (asymptomatic) infections fraction together with the rate of transmission of the undocumented infections. They estimate the undocumented rate to be 86%. Engbert et al. [13] use an EnKF for regional transmission modeling. They propose estimating time-independent parameters by maximizing the likelihood in a stochastic SEIR model to capture the dynamics of the pandemic at regional levels. Evensen et al. [14] apply an ensemble Kalman smoother technique to a meta-population model. The evolution of epidemiological parameters is estimated over a long time period. The technique can capture the abrupt changes in the reproduction number found in several countries following the implementation of lockdown measures.

There is a strong dependence between the severity of COVID-19 symptoms and age. Infections among children and young people often result in asymptomatic cases. On the other hand, Individuals aged over 60 tend to develop the most severe symptoms, often necessitating hospitalization during the course of the illness. Transmission effects have also been associated with age [15–17]: while children under 10 years old appear to have a low susceptibility to infection, people over 60 are highly susceptible. Hence, a technique for real time monitoring must use age-disaggregated data for an effective response to epidemics [18,19]. This includes accurate prediction of hospital beds and ventilators availability. Moreover, identifying age-dependent patterns in virus transmission is essential for policymaking regarding non-pharmaceutical interventions, such as deciding when to open or close schools [14].

Estimating the number of contacts between individuals for a particular population poses a challenge. This can be achieved by statistically significant population surveys. Arregui et al. [20] use surveys from eight countries [21] to extrapolate known contact matrices to other countries. Klepac et al. [22] use the data collected from a smartphone application in the UK to infer social interactions. The data contains the contact history of each user labeled by age groups, so that an empirical statistical contact matrix of the population is estimated. This matrix was then utilized in an agent-based model (ABM) to simulate an influenza-like outbreak, contributing to the BBC documentary *Contagion*. These works use a fixed contact matrix to study the evolution of epidemics and there is no estimation of time-varying contact rates.

Previous works assume a time-independent transmission matrix, however the response to COVID evolution showed that the social distancing measures changed with time and independently for different age groups. Motivated by this limitation, this work explores alternatives to a fixed transmission matrix by proposing and estimating time-varying transmission matrix parameterizations. We aim to estimate changes over time in the transmission matrix, including variations in mobility within specific age groups. To achieve this, we integrate a meta-population SEIRHD model with a stochastic EnKF to assimilate age-structured cumulative cases and deaths. Alongside the transmission parameters, we also estimate other crucial parameters, such as the effective reproduction number and the fraction of detected cases and deaths, utilizing information on the age-structured data related to the spread of the virus.

The outline of this article is as follows:

- In Section 2 we show our model and introduce the data assimilation framework.
- In Section 3 we give details of the real-world data utilized, present the general experimental details and show the different contact matrix parameterizations used.
- In Section 4 we present and discuss the results, each subsection corresponds to a different experiment including synthetic and real-world data experiments.
- In Section 5 we draw the conclusions of our investigation.

## 2 Technique details

### 2.1 Compartmental epidemiological model

In this work, the evolution of COVID-19 is modeled for the entire population of a region, which is assumed to be isolated. The model is an extension of a basic SEIR model [23], applied to a closed population (i.e. no births, deaths, immigration or emigration) divided into $n$ age groups. This model classifies individuals of a population into the following mutually exclusive categories: $S_j$ (susceptible), $E_j$ (exposed but not infectious), $I_j$ (infected), $M_j$ (mild symptoms), $T_j$ (severe symptoms), $C_j$ (critical symptoms), $R_j$ (recovered) and $D_j$ (dead). The index $j = 1, \ldots, n$ denotes the corresponding age group.

The flow between epidemiological categories of the model is shown in Fig 1. Infected individuals $I_j$ in the age group $j$ can interact with susceptible individuals $S_k$ in the age group $k$ with a transmission rate $\lambda_{jk}$. Susceptible individuals $S_j$ exposed to the disease move to the exposed compartment $E_j$. Individuals in this compartment do not transmit the virus. After a mean incubation time $\tau^E$, exposed individuals move to the infected group $I_j$. At this stage, individuals can spread the virus to susceptible persons during the period $\tau^I$. Subsequently, individuals move to the compartments $T_j$, $C_j$ or $M_j$ with probabilities $f_j^T, f_j^C$ and $1 - f_j^T - f_j^C$, respectively. The group $T_j$ comprises individuals with severe cases that require hospitalization and, after a time $\tau^T$, recover from the disease and move to the recovered compartment $R_j$. The compartment $C_j$ (critical) represents the individuals with severe cases that require hospitalizations and, after a time $\tau^C$, die and move to the dead compartment $D_j$. The compartment $M_j$ consists of the individuals who present mild symptoms and require no hospitalization, and after a time $\tau^M$, they transit to the recovered compartment. The model does not incorporate transitions between $M_j$, $T_j$ and $C_j$, because these compartments do not represent the illness progression but rather indicate the worst state that an individual has reached. If an individual is misclassified in one of these compartments, the data assimilation system will make the required adjustments to correct it, as explained in the next section. After a period $\tau^R$, individuals from the recovered compartment become susceptible again, given that SARS-COV-2 immunity diminishes substantially after 5-7 months [24]. The compartments are designed to characterize the dynamics of COVID-19 infection. Individuals are unable to transmit the virus in the initial incubation phase becoming infectious afterward. They are also expected to be isolated once the symptoms are apparent (or tested positive). Therefore, individuals in $M_j$, $T_j$, or $C_j$ are expected to be isolated and do not spread the disease, only individuals in the compartment $I_j$ do.

The model parameters are the transmission matrix parameters $\lambda_{jk}$ (which is the number of contacts that a person in group $j$ have with persons in group $k$, in a period of

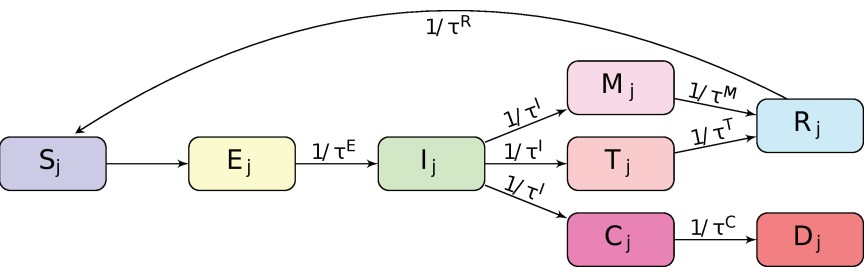

**Fig 1. Diagram of the compartmental model.** An individual moves to the next compartment after a period $\tau^{\mathcal{X}}$, which depends on the compartment $\mathcal{X}$.

time $\Delta t$, multiplied by the probability of that contact resulting in an infection), the average time an individual stays in each of the epidemiological states $\tau^E$, $\tau^I$, $\tau^M$, $\tau^T$, $\tau^C$ and $\tau^R$, and the fractions of infections $f_j^T$ and $f_j^C$ of $I_j$ moving to $T_j$ and $C_j$, respectively. The population of each age compartment is constrained by the total population of the age group $N_j$.

The resulting model equations are

$$
\begin{aligned}
\frac{\partial S_j}{\partial t} &= -\frac{S_j}{\tau^I N_j} \sum_{k=1}^{n} \lambda_{jk} I_k + \frac{R_j}{\tau^R} \\
\frac{\partial E_j}{\partial t} &= \frac{S_j}{\tau^I N_j} \sum_{k=1}^{n} \lambda_{jk} I_k - \frac{E_j}{\tau^E} \\
\frac{\partial I_j}{\partial t} &= \frac{E_j}{\tau^E} - \frac{I_j}{\tau^I} \\
\frac{\partial T_j}{\partial t} &= f_j^T \frac{I_j}{\tau^I} - \frac{T_j}{\tau^T} \\
\frac{\partial C_j}{\partial t} &= f_j^C \frac{I_j}{\tau^I} - \frac{C_j}{\tau^C} \\
\frac{\partial M_j}{\partial t} &= \left(1 - f_j^T - f_j^C\right)\frac{I_j}{\tau^I} - \frac{M_j}{\tau^M} \\
\frac{\partial D_j}{\partial t} &= \frac{T_j}{\tau^C} \\
\frac{\partial R_j}{\partial t} &= \frac{M_j}{\tau^M} + \frac{T_j}{\tau^T} - \frac{R_j}{\tau^R} \\
N_j &= S_j + E_j + I_j + M_j + T_j + C_j + R_j + D_j
\end{aligned}
\tag{1}
$$

Table 1 summarizes the variables and the parameters and Table 2 shows the numeric values of all the fixed parameters. In [25], $\tau^E = 4$ d is reported, and in [26] $\tau^I = 5$ d, while $\tau^T$ and $\tau^C$ are set both to 15 days following [14]. The fractions of hospitalizations, $f^T$, in Table 2 were estimated from the early stages of the pandemic using the available data. Eqs (1) are integrated with the Euler method using a time step of 1 h.

The most important parameters controlling the spread of a disease in a meta-population model are the elements of the transmission matrix. In a population divided into age-compartments, these elements represent the interaction between the infected and susceptible age groups, hence they are the main driver of the disease evolution. One of the central objectives of this work is to parameterize and estimate, from observed data, the transmission matrix from observations to obtain a better representation of the propagation between age groups. In Eq (1), the elements of the transition matrix are not independent [20]: the total number of contacts that individuals in the group $j$ have with those in group $k$ has to be equal to the total amount of contacts that individuals in group $k$ have with those in group $j$:

$$
\lambda_{jk} N_j = \lambda_{kj} N_k.
\tag{2}
$$

The most relevant parameter in epidemiological modeling is the basic reproduction number. It represents the mean number of new infected individuals caused by one infected person in a totally susceptible population. The basic reproduction number may be estimated in compartmental models by linearizing the dynamics of the infected differential subsystem, which is the part of the model that governs the production of new infections when all individuals are susceptible (in a SEIR model, for example, this subsystem are the compartments

**Table 1. Model variables and parameters.**

| Variables | |
|---|---|
| $S_j$ | Susceptible individuals |
| $E_j$ | Exposed individuals (non-contagious yet) |
| $I_j$ | Infected individuals (contagious) |
| $M_j$ | Infected with mild symptoms (isolated) |
| $T_j$ | Individuals with severe symptoms that eventually will recovered (isolated) |
| $C_j$ | Individuals with critical symptoms that eventually will die (isolated) |
| $R_j$ | Recovered |
| $D_j$ | Dead |
| Parameters | |
| $\lambda_{jk}$ | Transmission rate between the age group k to j |
| $\tau^E$ | Incubation period. |
| $\tau^I$ | Infection period. |
| $\tau^M$ | Recovery period for mild infections. |
| $\tau^T$ | Recovery period for severe infections. |
| $\tau^C$ | Time until death. |
| $\tau^R$ | Time until immunity vanishes. |
| $f_j^T$ | Fraction of the infected individuals in the age group $j$ that develops severe symptoms. |
| $f_j^C$ | Fraction of the infected individuals in the age group $j$ that eventually dies. |
| $N_j$ | Total number of individuals in the age group j. |

**Table 2. Numeric values of the model parameters. All time scales ($\tau$'s) are expressed in days.**

| age group-dependent parameters | | | |
|---|---|---|---|
| age group | 1 | 2 | 3 |
| Age range | 0-29 | 30-64 | 65-103 |
| $f^T$ | 0.01 | 0.05 | 0.26 |
| global parameters | | | |
| $\tau^E$ | 4 | | |
| $\tau^I$ | 5 | | |
| $\tau^M$ | 7 | | |
| $\tau^T$ | 15 | | |
| $\tau^C$ | 15 | | |
| $\tau^R$ | 150 | | |

SEI). The resulting Jacobian matrix is known as *next generation matrix* [27], whose spectral radius corresponds to the basic reproduction number $R_0$. If the linearization of the infected subsystem is conducted at time *t*, the spectral radius of the resulting matrix is known as the effective reproduction number $R_{\text{eff}}$. This represents the number of secondary cases that an infected individual produces at time *t*, assuming the remaining non-infected or recovered population is susceptible. A review of the topic can be found in [28].

## 2.2 State-parameter estimation with ensemble-based data assimilation

The evolution of epidemiological variables can be modeled as a partially observed time-evolving process, i.e. a hidden Markov model [29]. Within this framework, the evolution of the state of the system can be written as

$$\mathbf{x}_{k+1} = \mathcal{M}(\mathbf{x}_k) + \eta_k, \tag{3}$$

where $\mathbf{x}_k$ is the state of the system at time $k$, $\mathcal{M}()$ is the dynamical model and $\eta_k$ is the model error, representing simplifications and numerical approximations in the model. It is assumed to be a realization of $\mathcal{N}(\mathbf{0}, \mathbf{Q})$ (normal distribution of mean $\mathbf{0}$ and covariance matrix $\mathbf{Q}$). In our case, the dynamical model $\mathcal{M}()$ is (1). The second equation set forming the hidden Markov model corresponds to the observational map (in other words, the relationship between state variables and the observed quantities that can be measured such as the daily number of new infections). The observations $\mathbf{y}_k$ are related to the state $\mathbf{x}_k$ by the observation operator $\mathcal{H}_k$ which maps the space of state variables to the observational space:

$$\mathbf{y}_k = \mathcal{H}_k(\mathbf{x}_k) + \epsilon_k, \tag{4}$$

where $\epsilon_k$ is the observation error assumed to be a realization of $\mathcal{N}(\mathbf{0}, \mathbf{R})$ and $\mathbf{R}$ is the observation covariance matrix (assumed known). This represents the uncertainty in the observed values such as those produced by limitations in testing strategies, delays in the transmission of reports, etc. The observation covariance matrix is assumed diagonal

$$\left[\mathbf{R}\right]_{ii} = R_i \tag{5}$$

where $R_i$ is the variance of the observed variables. In Section 3, we define $R_i$. The observation covariance may be estimated with expectation-maximization [30], however here we consider a fixed observation covariance.

If the the hidden Markov model (3) and (4) satisfies the conditions of detailed balance, the inference may be greatly simplified. Here we assume a general framework and use filtering theory for the inference.

In filtering theory, the estimation problem involves obtaining the conditional probability density function (pdf) of $\mathbf{x}_k$ knowing the current and past observations $\mathbf{Y}_k = (\mathbf{y}_1, \mathbf{y}_2, ..., \mathbf{y}_k)$, denoted by $p(\mathbf{x}_k | \mathbf{Y}_k)$ (a.k.a. filtering or analysis distribution). We can obtain the prediction pdf by performing a forecast step

$$p(\mathbf{x}_k | \mathbf{Y}_{k-1}) = \int d\mathbf{x}_{k-1}\, p(\mathbf{x}_k | \mathbf{x}_{k-1})\, p(\mathbf{x}_{k-1} | \mathbf{Y}_{k-1}) \tag{6}$$

then, using Bayes theorem, the posterior density conditioned on the set of observations is derived,

$$p(\mathbf{x}_k | \mathbf{Y}_k) = \frac{p(\mathbf{y}_k | \mathbf{x}_k) p(\mathbf{x}_k | \mathbf{Y}_{k-1})}{\int d\mathbf{x}_k p(\mathbf{y}_k | \mathbf{x}_k) p(\mathbf{x}_k | \mathbf{Y}_{k-1})}. \tag{7}$$

Eqs. (6) and (7) can be solved sequentially every time new observations $\mathbf{y}_k$ are available, but they have to be integrated over the entire state space, which is usually computationally intractable. However, using a sample-based representation of the distributions, the forecast step can be approximated by a Monte Carlo approach by simply evolving every sample point forward with the model $\mathcal{M}()$. In this work we employ the EnKF, which is a Monte Carlo non-linear extension of the Kalman Filter [31]. The analysis distribution is represented by an ensemble of possible states. The resulting analysis state members are of the form

$$\mathbf{x}_k^{\mathrm{a},(i)} = \mathbf{x}_k^{\mathrm{f},(i)} + \mathbf{K}_k \left( \mathbf{y}_k^{(i)} - \mathcal{H}(\mathbf{x}_k^{\mathrm{f},(i)}) \right) \tag{8}$$

where the supra-index $i$ denotes the $i$-th ensemble member. Each forecast state member $\mathbf{x}_k^{\mathrm{f},(i)}$ is obtained by evolving the model (1) from the previous analysis state: $\mathbf{x}_k^{\mathrm{f},(i)} = \mathcal{M}(\mathbf{x}_{k-1}^{\mathrm{a},(i)})$, and

then corrected with an innovation term based on the difference between the observation and the forecast state. In Eq (8), the observation vector is perturbed with Gaussian noise: $\mathbf{y}_k^{(i)} = \mathbf{y}_k + \mu_k^{(i)}$, where $\mu_k^{(i)} \sim N(\mathbf{0}, \mathbf{R})$. This is required to obtain a sample covariance of the analysis state members with the expected analysis covariance [32]. The matrix $\mathbf{K}_k$, referred to as *Kalman gain*, gives the weight to the innovation term, and is the rate between the forecast uncertainty and the total uncertainty, namely

$$\mathbf{K}_k = \mathbf{P}_k^{\mathrm{f}} \mathbf{H}^\top \left( \mathbf{H} \mathbf{P}_k^f \mathbf{H}^\top + \mathbf{R} \right)^{-1} \tag{9}$$

where $\mathbf{P}_k^f$ is the forecast error covariance and $\mathbf{H}$ is the tangent linear observational operator defined by

$$H_{ij} = \left. \frac{\partial \mathcal{H}_i}{\partial x_j} \right|_{\mathbf{x}_k} \tag{10}$$

In an ensemble Kalman filter, $\mathbf{P}_k^f$ is estimated from the ensemble of forecasted state vectors at time $k$:

$$\mathbf{P}_k^{\mathrm{f}} = \frac{1}{m-1} \sum_{i=1}^m \left( \mathbf{x}_k^{\mathrm{f},(i)} - \overline{\mathbf{x}}_k^{\mathrm{f}} \right) \left( \mathbf{x}_k^{\mathrm{f},(i)} - \overline{\mathbf{x}}_k^{\mathrm{f}} \right)^\top , \qquad \overline{\mathbf{x}}_k^{\mathrm{f}} = \frac{1}{m} \sum_{i=1}^m \mathbf{x}_k^{\mathrm{f},(i)} \tag{11}$$

The analysis mean state, $\overline{\mathbf{x}}_k^{\mathrm{a}} = \frac{1}{m} \sum_{i=1}^m \mathbf{x}_k^{\mathrm{a},(i)}$, provides a point estimate of the state of the system.

## 2.3 Observation operator

Observations are assumed to be cumulative cases ($y_j^c$) and deaths ($y_j^d$) both disaggregated by ages. The map from state to observation space is as follows: we assume that the cases are partially documented. This is achieved with a time-dependent parameter $0 < \gamma_j < 1$ in the observational operator, $\mathcal{H}$, which accounts for the sub-detection of cases. In other words, we assume there is a sub-detection bias in some observational variables. This parameter depends on the age group, since the symptoms may increase with age so that the amount of undocumented cases is larger for children. In the age group $j$, the relation between the cumulative observed cases ($y_j^c$) and observed deaths ($y_j^d$) and the state variables at time $k$ is

$$\mathbf{y} = \mathbf{H}\mathbf{x} + \epsilon, \tag{12}$$

where $\mathbf{y} = (y_j^c, y_j^d)^\top$, $\mathbf{x} = (S_j, E_j, I_j, M_j, T_j, C_j, R_j, D_j)^\top$ and

$$\mathbf{H} = \begin{pmatrix} 0 & 0 & \gamma_j & \gamma_j & \gamma_j & \gamma_j & \gamma_j & \gamma_j \\ 0 & 0 & 0 & 0 & 0 & 0 & 0 & 1 \end{pmatrix} \tag{13}$$

To avoid index overclutter, we do not include temporal index $k$ in this equation, but note that $\gamma$ is a time-varying parameter.

For parameter estimation with the ensemble-based data assimilation technique, the model parameters $\theta$ to be estimated and the state variables $S_j, E_j, I_j, M_j, T_j, C_j, R_j, D_j$ are concatenated together into an augmented state vector $\mathbf{x}$. Then, the model parameters are estimated in the same way as the state variables, using the EnKF. This parameter estimation methodology is known as *augmented state*. A review of parameter estimation using various data assimilation methods based on the state augmentation approach can be found in [33]. The fractions of detected cases $\gamma_j$ are also estimated in this way. The prior density at the initial time is

assumed Gaussian for both model variables and parameters. This is coherent with the Gaussian assumption of the Kalman filter. Although the parameters are not part of the model equations, their estimation can be conducted in the same way as for the model variables. They are estimated through their correlations with the observed variables. Therefore, parameter estimation depends crucially on an accurate quantification of the augmented forecast covariance matrix (11).

While chaotic dynamics drive the evolution of state variables leading to an increase in their ensemble spread, persistence is assumed for the time evolution of the parameters. Because of this, an inflation method is required to prevent the parameter ensemble spread from collapsing during the recurrence (e.g. Ruiz et al. 2013) [33].

We conducted preliminary experiments to evaluate the use of multiplicative inflation in the EnKF framework. Despite we use two independent inflation factors, one for the parameters and one for the state variables [34], we were not able to find a suitable set of inflation factors. The attempted combinations resulted in either filter divergence or poor estimation performance. Consequently, we opted for the stochastic approach originally proposed in [35]. The parameter evolution of each ensemble member is modeled as an independent autoregressive process or correlated random walk with correlation $\rho$ and standard deviation $\sigma$, this is

$$\theta_{k+1}^{(i)} = \bar{\theta}_k + \rho \, (\theta_k^{(i)} - \bar{\theta}_k) + \sigma \sqrt{1 - \rho^2} \, \xi, \tag{14}$$

where $\xi$ is a random vector sampled from a Gaussian distribution with zero mean and identity covariance matrix. The random perturbation is added before the analysis step and is only applied to the parameters $\theta_k^{(i)}$; no inflation is applied to the model state variables.

During the analysis update, the EnKF can result in non-physical values for some model parameters and ensemble members (e.g. negative values for the transmission matrix elements). This is a consequence of the assumption of Gaussian forecast error in the EnKF. To avoid this complication, we force the lower limit of all the estimated parameters to 0 in each ensemble member.

To summarize our estimation method, the EnKF methodology is represented concisely in Algorithm 1.

**Algorithm 1. Stochastic ensemble Kalman Filter**

**Require: H, R,** $\mathcal{M}()$, $\mathbf{y}_k$ and $\mathbf{x}_0^{a,(i)}$, i=1,...,m        ▷ Inputs and
                                              ensemble initialization
**do** $k = 1, 2, \ldots$                                   ▷ loop over time
        $\mathbf{x}_k^{f,(i)} = \mathcal{M}\,(\mathbf{x}_{k-1}^{a,(i)})$                                ▷ Forecast
        $\mathbf{P}_k^f = \frac{1}{m-1} \sum_{i=1}^{m} (\mathbf{x}_k^{f,(i)} - \overline{\mathbf{x}}_k^f)\,(\mathbf{x}_k^{f,(i)} - \overline{\mathbf{x}}_k^f)^\top$        ▷ forecast covariance
        $\mathbf{K}_k = \mathbf{P}_k^f \mathbf{H}^\top \,(\mathbf{H}\mathbf{P}_k^f\mathbf{H}^\top + \mathbf{R})^{-1}$                        ▷ Kalman gain
        $\mathbf{y}_k^{(i)} = \mathbf{y}_k + \mu^{(i)},$            ▷ Data input as perturbed observations
        $\theta_k^{(i)} = \bar{\theta}_k + \rho\,(\theta_k^{(i)} - \bar{\theta}_k) + \sigma\sqrt{1 - \rho^2}\,\xi$            ▷ Inflate parameters and
augment state
        $\mathbf{x}_k^{a,(i)} = \mathbf{x}_k^f + \mathbf{K}_k\,(\mathbf{y}_k^{(i)} - \mathbf{h}\,(\mathbf{x}_k^{f,(i)}))$                        ▷ Analysis
**end do**

## 3 Experimental details

### 3.1 Transmission matrix parameterizations

For an $n \times n$ transmission matrix there are $\frac{n^2+n}{2}$ independent parameters to be estimated instead of $n^2$ because of the restriction (2). In our case we use three age groups, so the resulting transmission matrix is

$$\Lambda = \begin{pmatrix} \lambda_{11} & \lambda_{12} & \lambda_{13} \\ \frac{N_1}{N_2}\lambda_{12} & \lambda_{22} & \lambda_{23} \\ \frac{N_1}{N_3}\lambda_{13} & \frac{N_2}{N_3}\lambda_{23} & \lambda_{33} \end{pmatrix} \qquad (15)$$

where parameters $\lambda_{ij}$ depend on time.

As shown in the experiments in Section 4, the parameters of (15) are not all identifiable when only information of the accumulated infection cases in each group is available, without details regarding the specific age group responsible for the new exposed individuals.

To overcome this limitation, we propose a parameterization for the transmission matrix with fewer parameters:

$$\Lambda = \begin{pmatrix} \lambda_{11} & \alpha\sqrt{\lambda_{11}\lambda_{22}} & \alpha\sqrt{\lambda_{11}\lambda_{33}} \\ \frac{N_1}{N_2}\alpha\sqrt{\lambda_{22}\lambda_{11}} & \lambda_{22} & \alpha\sqrt{\lambda_{22}\lambda_{33}} \\ \frac{N_1}{N_3}\alpha\sqrt{\lambda_{33}\lambda_{11}} & \frac{N_2}{N_3}\alpha\sqrt{\lambda_{33}\lambda_{22}} & \lambda_{33} \end{pmatrix} \qquad (16)$$

from now on, we call this matrix the parameterized transmission matrix.

This parameterization is a particular case of (15) where the upper diagonal parameters $ij$ are defined as a function of the diagonal elements of the row $i$ and column $j$: $\lambda_{ij} = \sqrt{\lambda_{ii}\lambda_{jj}}$, and the lower diagonal parameters are defined by the constraint (2). The parameter $\alpha$ controls the relative importance of inter-age group and intra-age group infections, with lower values giving more weight to the latter.

### 3.2 Data

We use three age groups in the range of $[0, 30)$, $[30, 65)$ and $[65, -]$ years. This division is motivated because we want to represent age groups with different activities, so that children and young individuals' activities are mainly school and universities, adults are the working age group and the senior population is assumed to be mainly retired. At the same time, these groups grossly represent different health profiles, with the senior population being the ones that most likely will develop severe symptoms, while the first age group are more likely to have minor symptoms. The total population is assumed to be 44.8 million divided into the three age groups by $2.2 \times 10^7$, $1.8 \times 10^7$, and $4.8 \times 10^6$, which represent the approximate number of people within the aforementioned age groups in Argentina (taken from the 2010 population census).

**3.2.1 Synthetic observations.** In a data assimilation context, a "twin" experiment is an idealized simulation in which we use a known epidemiological model with a set of "true" parameters to generate synthetic observations by first computing the model state evolution, and then adding random noise in the observation space to simulate the observation error (4). In this way, we generate cumulative cases and deaths for each age group using (1) and add random normal noise to them to generate the synthetic observations. Then, the EnKF is used to reconstruct the evolution of the system and the parameters from the syntethic observations. In this way, we can assess the estimated parameters given by the technique comparing them to the "true" parameters (i.e., the ones used to generate the synthetic observations). The

objective of the twin experiments is to evaluate the data assimilation-based parameter estimation in a context in which the true parameters are known and errors in the estimation can be accurately computed. We refer to a "true" state variable or parameter to the state variable or parameter used in the model to generate the synthetic observations.

The model used to simulate the observations uses a "true" transmission matrix which has the form (15).

The "true" parameters $\lambda_{ij} = \lambda^{True} = [\lambda_{11}, \lambda_{22}, \lambda_{33}, \lambda_{12}, \lambda_{13}, \lambda_{23}]$ are defined as

$$\lambda^{True} = \begin{cases} [1.6, 1.8, 1.4, 0.5, 0.4, 0.3] & \text{if } t \in [0, 80) \text{ d} \\ [0.4, 0.6, 0.3, 0.15, 0.13, 0.1] & \text{if } t \in [80, 140) \text{ d} \\ [1.6, 1.2, 1.35, 0.36, 0.25, 0.2] & \text{if } t \in [140, 300] \text{ d} \end{cases} \quad (17)$$

The decrease in the transmission matrix parameters at $t = 80$ d mimics the effect of a lockdown, and the increase at time 140 d represents a relaxation to normal conditions but with some sanitary measures (e.g. social distancing, mandatory use of masks in public spaces, etc). These conditions result in a double outbreak situation as observed in Argentina (and several other countries) in the first year of the pandemic.

This time-dependent "true" transmission matrix must be estimated by the assimilation in the twin experiment. Note that the relative changes in the parameters are different for different age groups (i.e. not proportional). We chose on purpose a transmission matrix that cannot be fully represented by the parameterization (16), so that the model used in the estimation is not perfect (some structural uncertainty is introduced in the parameterization process). Another motivation was to represent the resulting different levels of mobility that were found in different age groups.

The true values of the fraction of detected cases $\gamma_j$, $j = 1, 2, 3$, are taken to be 0.15, 0.2, and 0.3 corresponding to the young, adult, and senior age groups. For reference, a well-mixed population detection fraction was estimated in [12] to be 0.16 in the early stage of the pandemic. Intuitively, we expect a higher fraction of symptomatic cases for the elderly age group, as it is the most vulnerable population. The fraction of deaths $f_j^C$ of each age group is assumed to be 0.002, 0.05, and 0.1 . We used the values showed in [36], weighted by the corresponding population age group in Argentina.

Cumulative infected cases and deaths segmented by age groups are assumed to be daily observed during the time period. For the observational error, we set the observation standard deviation of the accumulated cases to $\sigma_c = \max(0.05 y_j^c, 100)$, where $y_j^c$ indicates the observed cumulative cases for every age group j. We assume that deaths are well documented so the standard deviation of the deaths observational error is $\sigma_d = \min(0.05 y_j^d, 5)$. The way we define the observational error means that eventually, all the observations will have the upper limit standard deviation after some time. These observation standard deviations are used to generate the synthetic observations and are also used as the observation variances $R_c, R_d = \sigma_c^2, \sigma_d^2$ in the ensemble Kalman filter (9).

**3.2.2 Real world observations.** For the real-world experiments, we use epidemiological data from Argentina collected by the National Health Surveillance System (SNVS, for its acronym in Spanish). The SNVS dataset is openly available (http://datos.salud.gob.ar/dataset/covid-19-casos-registrados-en-la-republica-argentina) and consists in all the reported tests from public and private tests. The available information for each case is, among other data, the date of the test, the province of residence, age, and whether the person required hospitalization (with or without respiratory support) and if he or she died during the infection. The first case of SARS-CoV-2 in Argentina was reported on March 3, 2020. Just 16 days later on March 19,

2020, a nationwide lockdown was established. The curated time series of the data used in this work are available at http://covid19.unne.edu.ar/obs_arg.csv.

# 4 Results

We present our results in the following order:

- In the subsection 4.1 we evaluate the model and data assimilation framework with synthetic observations (Section 3.2.1).
- In the subsection 4.2 we apply the methodology to COVID-19 data of Argentina (Section 3.2.2).
- In the subsection 4.3 we conduct forecasts to examine the performance of the meta-population model coupled with the EnKF using the data of Argentina.

## 4.1 Experiments with synthetic observations

The EnKF estimates all the variables of the system and the parameters of the transmission matrix, which are augmented to the system state vector. The dimension of the state vector is 24 (eight variables in each of the three age groups). From these, 21 variables are independent since there are 3 constraints (last Eq in (1)). The amount of estimated parameters is six in the case of the parameterized transmission matrix: three belonging to the parameterized transmission matrix and three corresponding to the fractions of detected cases, namely the parameter vector is $\theta = (\lambda_{11}, \lambda_{22}, \lambda_{33}, \gamma_1, \gamma_2, \gamma_3)$ so the augmented state vector dimension is 30. In the case of the full transmission matrix (15), there are nine estimated parameters: six from the matrix and three from the fraction of detected cases, i.e. $\theta = (\lambda_{11}, \lambda_{22}, \lambda_{33}, \lambda_{12}, \lambda_{13}, \lambda_{23}, \gamma_1, \gamma_2, \gamma_3)$ so that the augmented state vector dimension is 33.

As mentioned, the EnKF for parameter estimations requires an inflation approach for the parameter spread [33]. The filter exhibits convergence using the correlated random walk, (14), for high values of $\rho$ and $\sigma$ in the range $[0.001, 0.2]$. We performed preliminary experiments to determine optimal values by minimizing the root-mean-square error (RMSE, i.e. $[\sum_{j,k}(x_{jk}^a - x_{jk}^{\text{True}})]^{1/2}$) for the additive inflation parameter values, which we found to be $\sigma = 0.05$ and $\rho = 0.999$. The same random walk parameter values are used in the real data experiments (Section 4.2).

Fig 2 shows the estimated parameters for the synthetic observation experiment (Section 3.2.1) employing the full transmission matrix (15). There is some delay in the estimated transmission matrix parameters compared to the true parameters at the abrupt changes due to the lockdown measure (both in the beginning and end). Estimated parameters start to adjust to these abrupt changes a few days after the change and they converge to a new value 20-30 days later. The reason for this is that parameters in ensemble-based assimilation systems are estimated through the correlation with observed variables, so these state-parameter correlations take some cycles to adapt to abrupt changes. This behavior can be reduced by tuning up the amount of inflation, at the expense of having an increased spread in the ensemble of estimated parameters and state variables. Overall, the amplitude of the abrupt changes is rather well estimated beyond the mentioned delay.

To examine the identifiability and sensitivity to initial conditions of the estimated parameters, three independent experiments with different initial mean parameters (transmission matrix and fraction of detected cases) at $t = 0$ are shown in Fig 2 denoted as IC1, IC2 and IC3. Some of the time variability of the true parameters is captured. However, the different experiments converge to different estimated parameter values. Different initial conditions of

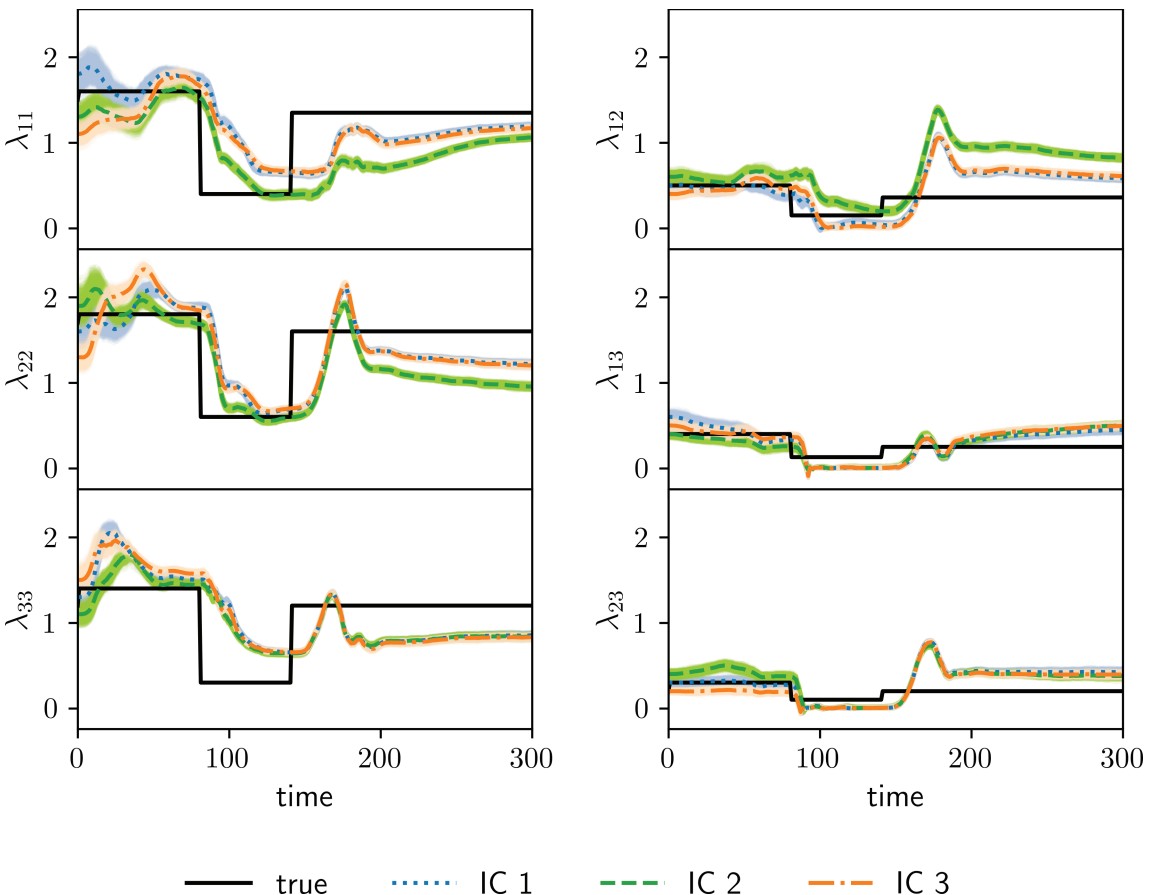

**Fig 2. Estimated diagonal values (left panels) and off-diagonal parameters (right panels) of the full transmission matrix are shown for the synthetic observation experiments using different initial conditions in the parameters (color lines, IC1, IC2 and IC3).** True parameter values are shown with black lines. Shading around the curves indicates the parameter spread.

the parameters result in different estimations, and neither of the three experiments is able to estimate precisely the true parameters (Fig 2).

The reason for this lack of identifiability is that an increase in the rate of cases say in the age group 1, can be attributed by the assimilation system to either a change in the parameter $\lambda_{12}$ or a change in $\lambda_{11}$ and $\lambda_{22}$. Both scenarios result in the same infection rates so the information provided by the observations is not enough to identify the actual scenario. In the experiment, $\lambda_{22}$ green curves in Fig 2 (IC2) present a greater underestimation at the end of the lockdown, compared to the orange and blue estimations (IC1 and IC3). This underestimation is balanced with the overestimation of $\lambda_{12}$, leading to an evolution of the number of cases consistent with the observations.

Fig 3 shows the estimated daily cases (left panels) and deaths (right panels) of the young (upper panels), adult (middle panels), and senior (lower panels) age groups, using the full transmission matrix (15). Although the assimilated observations are cumulative cases and deaths, we show daily cases, the new cases produced in one day, because differences are more visible. The term "daily cases" was taken from [12]. The three experiments with different initial conditions in the parameters give similar results (curves of the three experiments are indistinguishable in Fig 3). In the three experiments, the EnKF is able to keep track of the

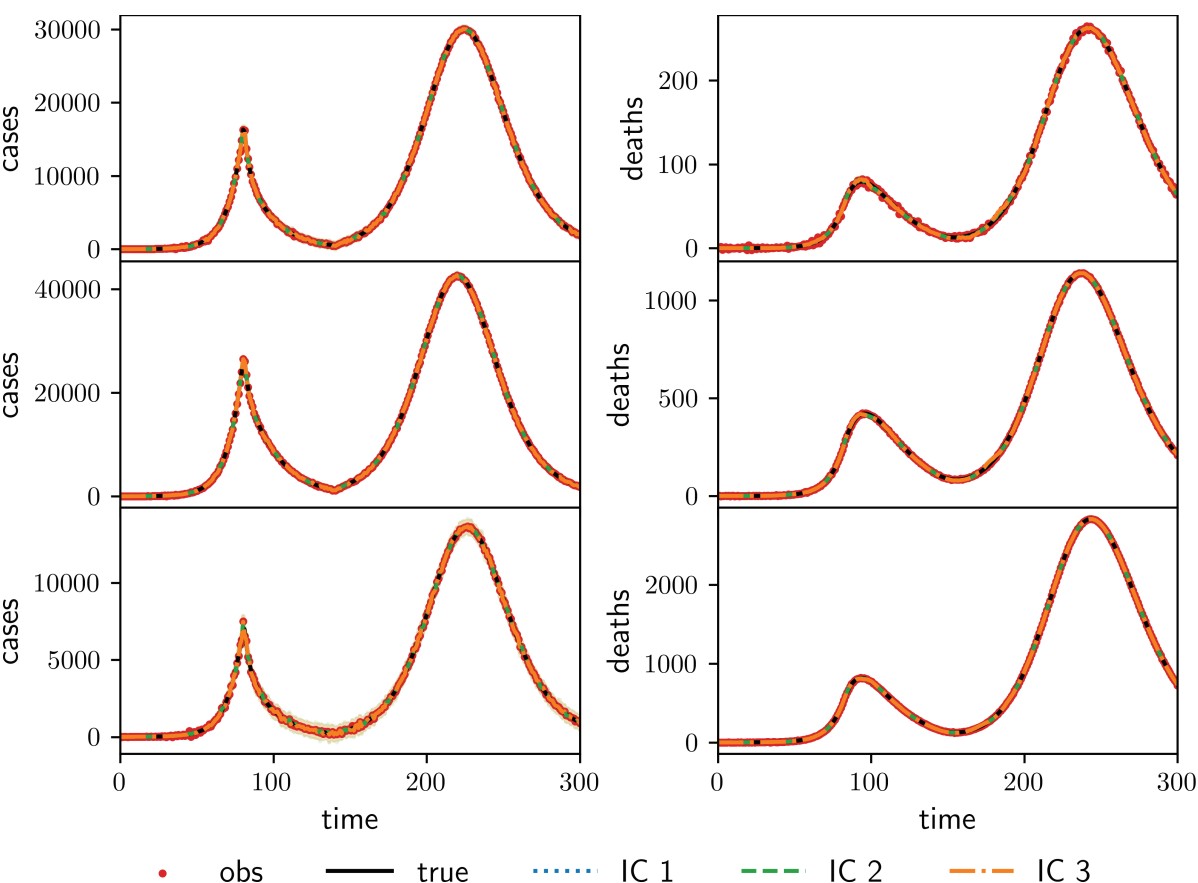

**Fig 3. Estimated daily cases (left panels) and daily deaths (right panels) of the young (upper panels), adult (middle panels), and senior (lower panels) age groups for the full transmission matrix experiment using synthetic observations using different initial conditions in the parameters (color lines, IC1, IC2 and IC3).** The observations are shown with red dots and black lines represent the true variable values (they are almost indistinguishable). Shaded areas around colored curves indicate the corresponding variable spread.

observations of cases and deaths in all the age groups, even though the transmission matrix parameters are not identifiable. The ensemble dispersion in the senior age group is relatively larger because the population is almost five times lower than in the other age groups, and all the age groups have the same observation error upper limit so that the relative error of the estimation is higher.

Given that the transmission matrix parameters are not identifiable using the matrix form (15), we conducted a second set of estimation experiments using the proposed parameterization (16) and the same set of synthetic observations. We took $\alpha = 0.25$ in (15), which represents the degree of intra-group contagious. Fig 4 shows the estimated parameters of the parameterized transmission matrix. The right panels show the values of the diagonal, and the left ones show the values of the upper off-diagonal.

The three experiments converge to the same estimated parameter values, independently of the initial condition. The true values of the parameters cannot be estimated precisely because this parameterization is not able to fit the structure of the prescribed true transmission matrix (17). The parameter estimates in Fig 4 also show a delay in the representation of the sudden

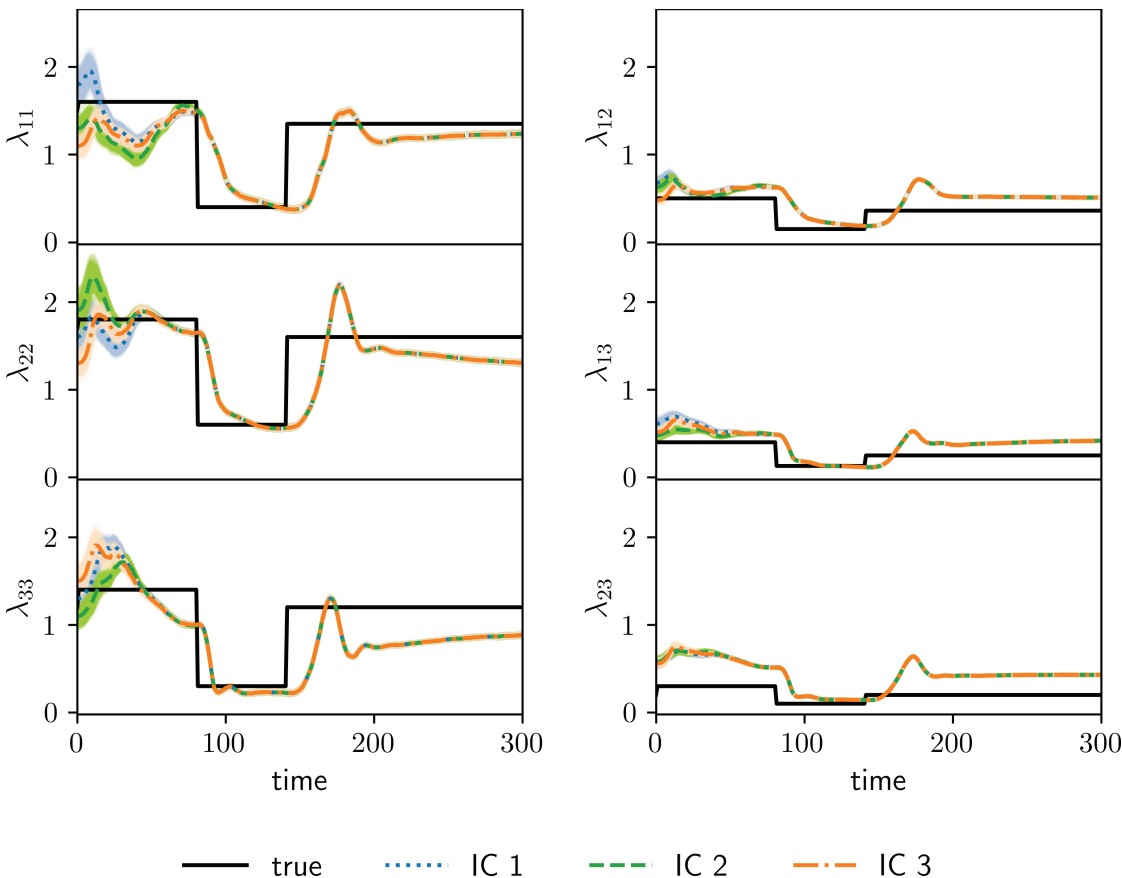

**Fig 4. Estimated diagonal values of the parameterized transmission matrix (left panels) and off-diagonal values (right panels) using synthetic observations for different initial conditions (colored lines, IC1, IC2 and IC3), and the true parameter values (black lines).** Shaded areas around colored curves indicate the parameter ensemble.

parameter changes found at the beginning and at the end of the lockdown period, as found in Fig 2.

Fig 5 shows the effective reproduction number calculated with the next generation matrix for the experiment corresponding to the parameterized transmission matrix (left panel) and to the full transmission matrix (right panel). The true values of $R_{\text{eff}}$ can be accurately estimated with both parameterizations (apart from the delay in parameter changes), even when the true transmission matrix is non-reproducible by the parameterized transmission matrix. This result suggests that our parameterized transmission matrix is sufficiently flexible to capture the system's $R_{\text{eff}}$ and its temporal evolution. At the same time, its low enough dimensionality ensures identifiability of its parameters from the available observations.

Fig 6 shows the fraction of detected cases of each age group (right panels). We expect these parameters to be correlated to the observed accumulated cases and deaths. Therefore, the system should be able to constrain them. The true values of $\gamma_j$ are accurately estimated by the assimilation system, regardless of the initial condition. The spurious peaks estimated in the parameterized transmission matrix at the lockdown transitions are also found in the $\gamma_j$ parameters around time 80 d and, with much less intensity, at 170 d.

In the previous experiment, we estimated a parameterized transmission matrix and the fraction of detected cases $\gamma_j$. The cases, deaths, and the parameters $\lambda_{jk}$ can also be estimated

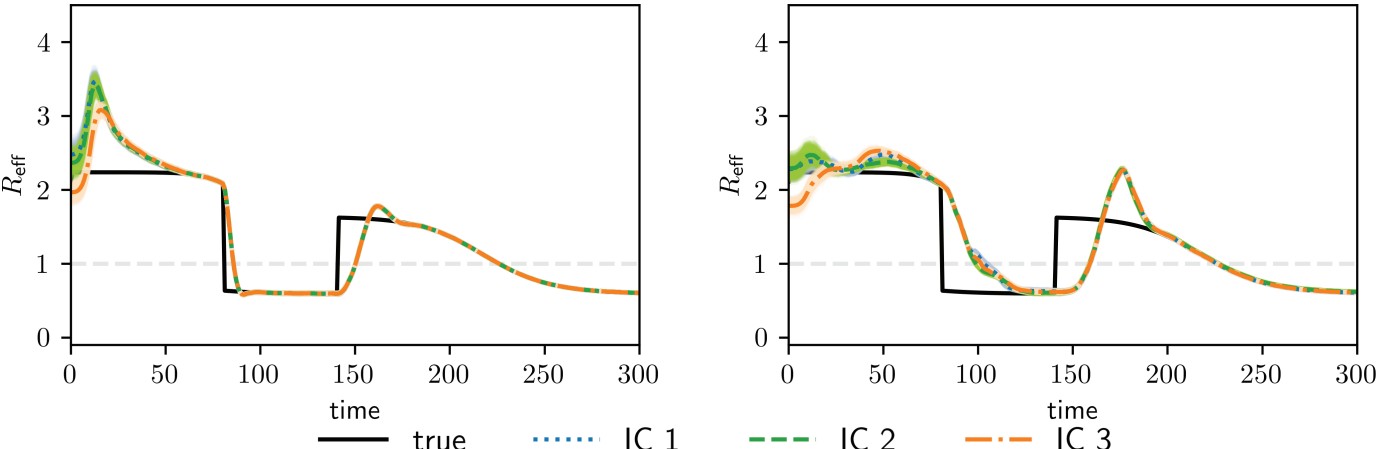

**Fig 5. Estimated effective reproduction number using the parameterized transmission matrix (left panel) and the full transmission matrix (right panel).** The two panels correspond to different assimilation experiments. Colored curves represent estimations with different initial conditions (IC1, IC2 and IC3), and black curves represent the true parameter values. Shading areas around colored curves indicate the parameter spread.

alongside the fractions of deaths $f_j^C$ instead of $\gamma_j$. To illustrate this, we fix $\gamma_j$ equal to the true values and perform three experiments that estimate the transmission matrix and the fraction of deaths. The parameters $\lambda_{jk}$ are similar to the ones shown in Fig 4. The obtained $f_j^C$ estimates are shown in the left panels of Fig 6. In all experiments, the estimated parameters converge to the true values, and the sudden change in the estimations is again observed at the times when the true transmission matrix parameters change. The shading in the upper right panel of Fig 6, parameter $f_1^C$, is limited to zero because this is the lower bound imposed to parameters in the different ensemble members.

## 4.2 Experiments with the Argentinian COVID-19 data

An experiment was conducted with the same assimilation system as in the previous section but using the real COVID-19 data from Argentina (Section 3.2.2). In contrast to the twin experiments, the observations in this non-synthetic case may be biased and the observation error covariance is unknown. Indeed, the observed cases are highly noisy. One of the sources of the noise is due to the fact that testing and reports diminish on weekends and increases on Mondays and Tuesdays due to delayed reports, resulting in an spurious weekly cycle in the observations.

We estimate the time-dependent fraction of deaths and the parameterized transmission matrix (16) with $\alpha = 0.25$ in the real observation experiments. The parameter vector in this case is $\theta = (\lambda_{11}, \lambda_{22}, \lambda_{33}, f_1^C, f_2^C, f_3^C)$. The estimation must account for a time interval of almost 1.5 years, time in which the SARS-Cov-2 virus underwent multiple mutations, changing the severity of the symptoms, there were advancements in treatments within the healthcare system and it includes the start of the vaccination period. This turns crucial the use of time-dependent parameters.

Fig 7 shows the daily cases (left panels) and daily deaths (right panels) of the young (top panels), adult (middle panels), and senior (bottom panels) age groups. The filter is able to keep track of the observations of each age group. As in the twin experiments, we use three sets of initial conditions, and they yield the same estimation of cases and deaths (in Fig 7 only IC3 estimates are visible). The high-frequency cycle found in the estimations of daily cases and

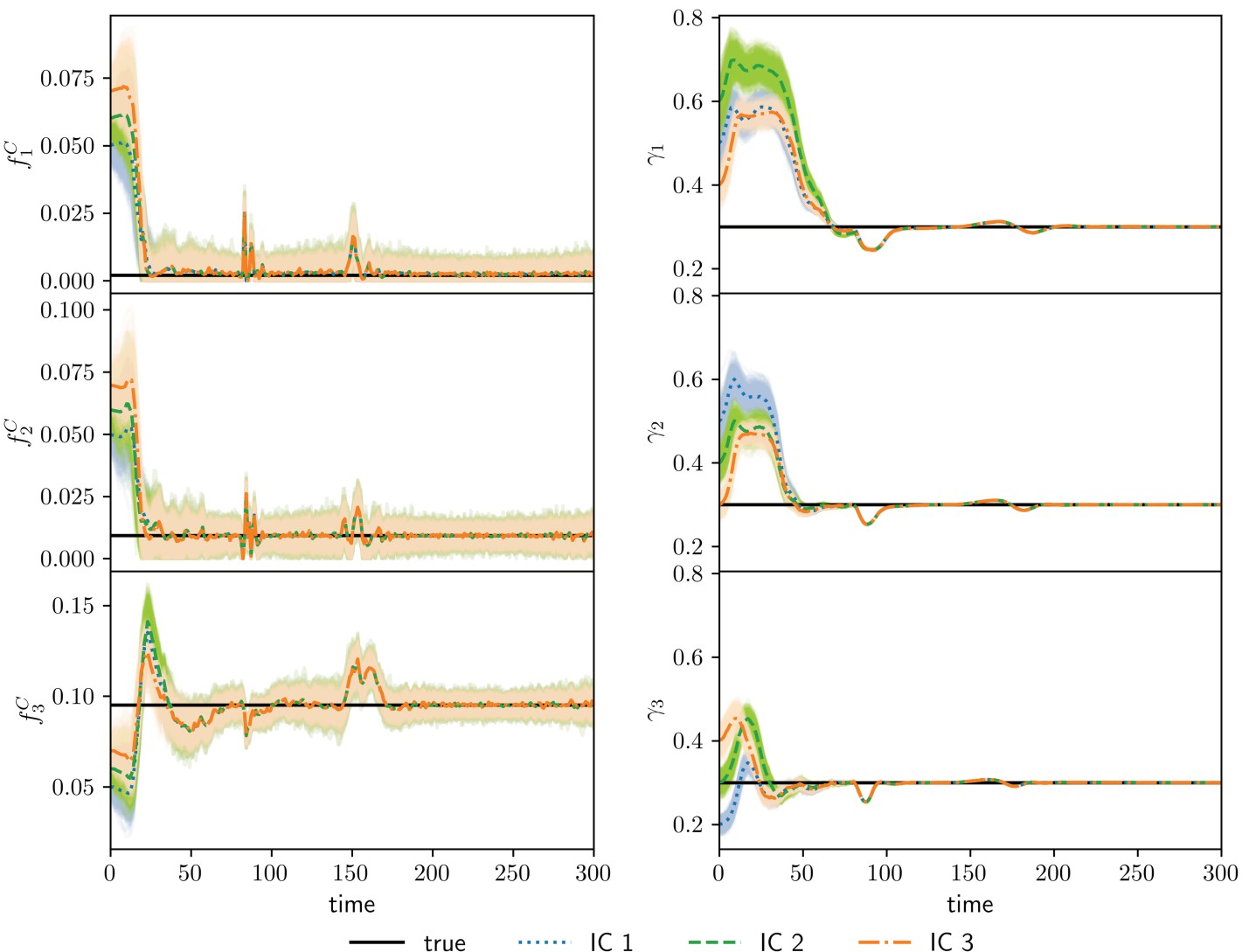

**Fig 6. Estimated fraction of deaths for each age group (left panels) and estimated fraction of detected cases for each age group (right panels).** Left and right panels correspond to different assimilation experiments (either detected fractions or fractions of deaths are estimated). Estimations with different initial conditions are shown with colored curves (IC1, IC2 and IC3), and black curves indicate the true parameter values. Shading around colored curves represent the parameter spread.

deaths corresponds to the above mentioned spurious weekly cycle. If required, this effect can be mitigated by increasing the observational error of the cases, at the expense of an increase of the uncertainty of the estimations.

The estimated deaths in the young age group have a high dispersion because of the relatively few observed cases compared to the other age groups and their relative errors. Although the estimated accumulated number of deaths is always positive, the daily changes in the number of deaths are sometimes negative for some ensemble members in the young compartment. This non-physical behavior is a consequence of the updates introduced by the observations which may eventually result in the reduction of the estimated number of deaths in order to better fit the observed values.

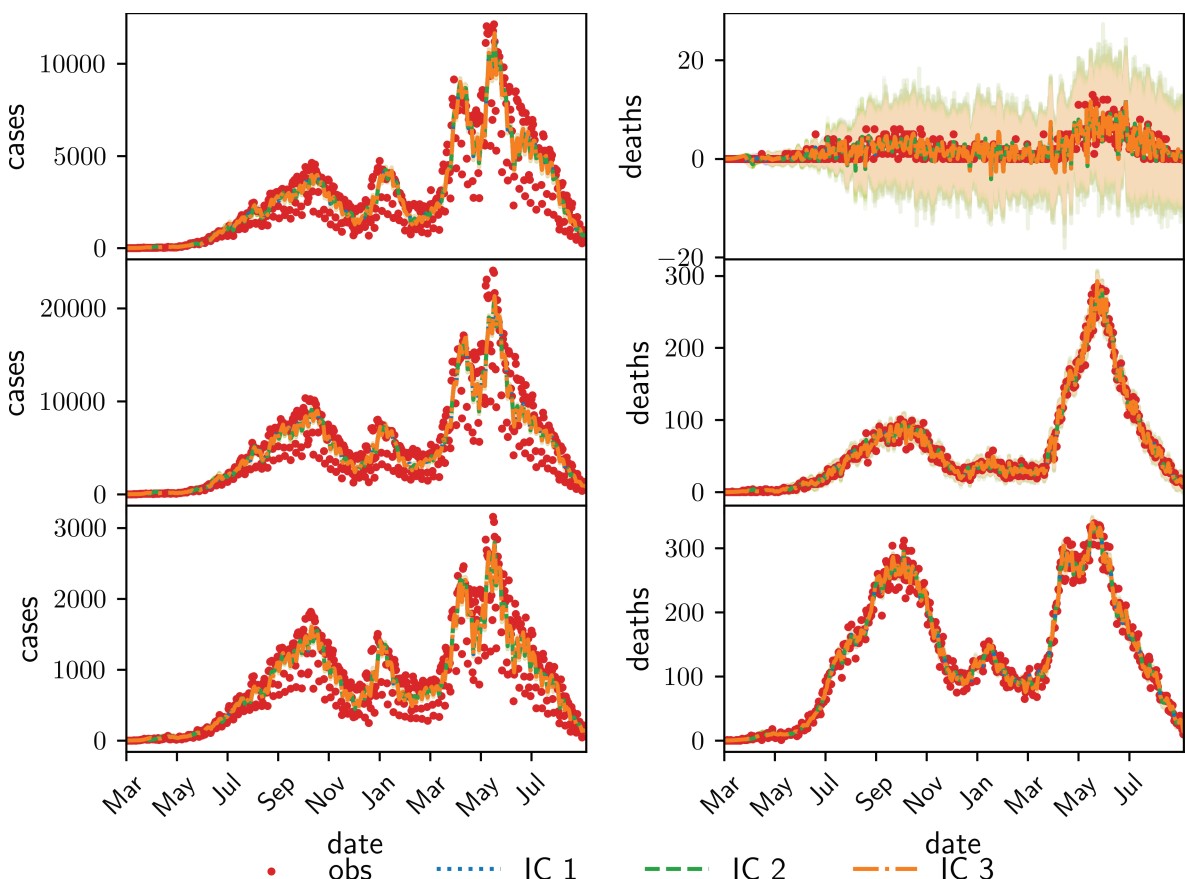

**Fig 7. Estimated daily cases (left-side panels) and deaths (right-side panels) using the parameterized transmission matrix in the Argentinian data experiments for the young (upper panels), adult (middle panels) and senior age groups (bottom panels).** Observations are in red dots. Shades around the curves represent the estimated variable uncertainty. The lines of the three estimations, IC1, IC2 and IC3, are indistinguishable.

Fig 8 shows the three independent parameters $\lambda_i$, $i = 1, 2, 3$ of the parameterized transmission matrix (left panels), and the upper off-diagonal parameters $\lambda_{ij} = \alpha\sqrt{\lambda_{ii}\lambda_{jj}}$ (right panels). All different initial conditions yield the same estimations of the parameters. There is a predominance of the parameters $\lambda_{11}$ and $\lambda_{22}$ given that the majority of the cases occur in the first two age groups. Consequently the interaction parameter $\lambda_{12}$ between young people and adults is higher compared to $\lambda_{13}$ and $\lambda_{23}$. The filter takes a couple of months to estimate parameters, with a spinup time longer than for the synthetic experiments, so that the estimates during the first two months depend on the chosen initial mean parameters. This parameter behaviour is consistent with the results of Sauer et al [37], where it is shown that an optimal set of parameters of a SEIR model can be estimated using the complete time-series of cumulative cases, but parameter identifiability problems occur when using only cumulative cases in the pre-peak interval. Besides this work compares the use of Markov Chain Monte Carlo and data assimilation methods for inference.

Fig 9 shows the estimated effective reproduction number $R_{\text{eff}}$. The estimated parameter does not depend on the chosen initial conditions. The peak at the start of the pandemic corresponds to the unrestricted spread of the virus until the lockdown starts on March 27, 2020. The filter takes some days to fully estimate the reproduction number up until the lockdown

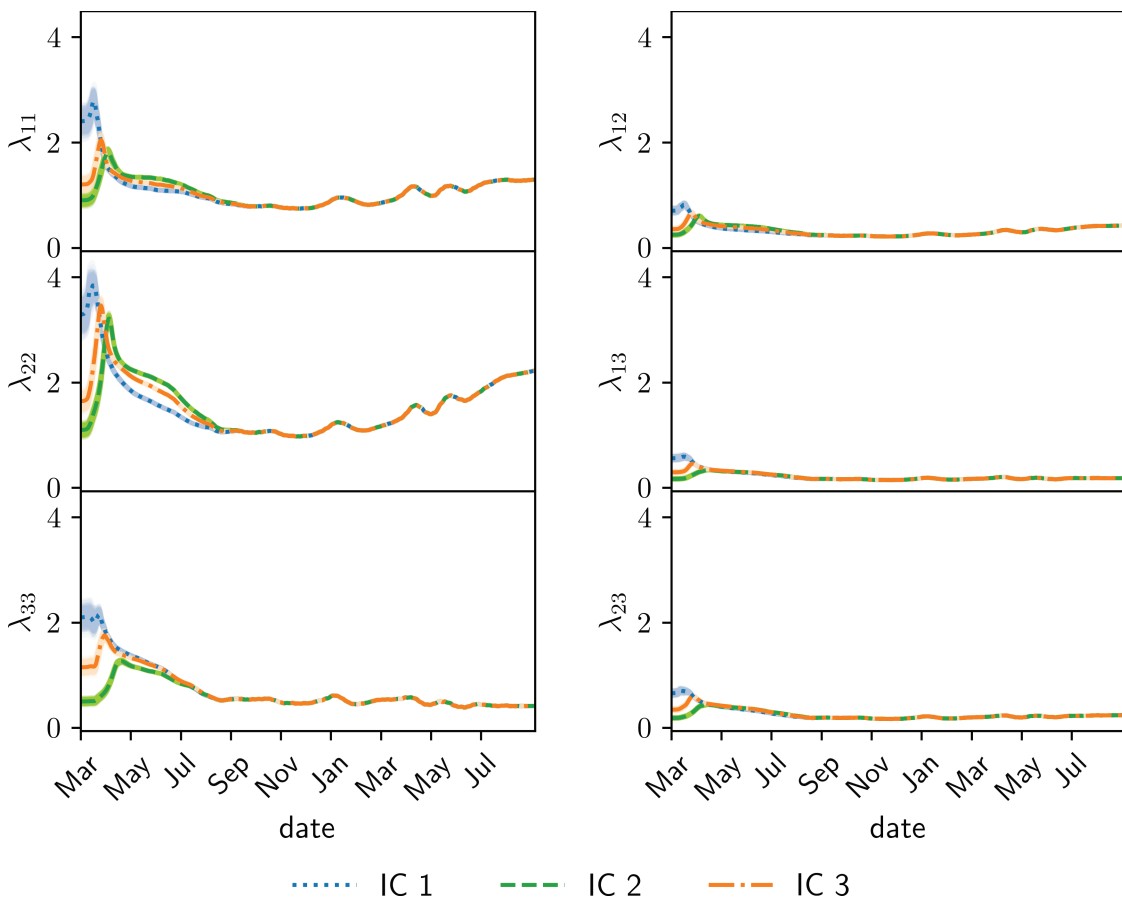

**Fig 8. Estimated diagonal values of the parameterized transmission matrix (left panels) and diagonal values (right panels) for different initial conditions (color lines IC1, IC2, IC3).** Shading around colored lines represent the parameter spread.

start given the inertia of the system, and after some days, it start to decreace. The estimated periods where $R_{eff} > 1$ corresponds to an increase in the cases up to the peaks. The initial condition IC1 (IC2) with a large (small) initial reproduction number reaches a higher (lower) peak on the lockdown date than the others. The lockdown start date is estimated earlier (later) than observed. The initial condition IC3 estimates correctly the time of the lockdown start. After this spinup time, all estimations converge to the same values.

Fig 10 shows the estimated fractions of deaths of the young (top plot), adult (middle plot) and senior (bottom plot) people age groups. The estimated parameters are slightly higher than the reference values 0.002, 0.05, and 0.1 [36] of the young, adult, and senior age groups. The death fraction of the senior age-group exhibits a large value in the first six months, about 0.175, but then it decreases substantially below 0.1. This may be caused by an improvement in the healthcare system: more hospital beds and artificial ventilators, early detection of low blood oxygen, to name a few.

### 4.3 Forecasts

To evaluate the potential use of the estimated parameters for decision making, we conducted an evaluation of the performance of the resulting forecasts using the estimated parameterized transmission matrix on the Argentinian COVID-19 data. Once assimilation of the last

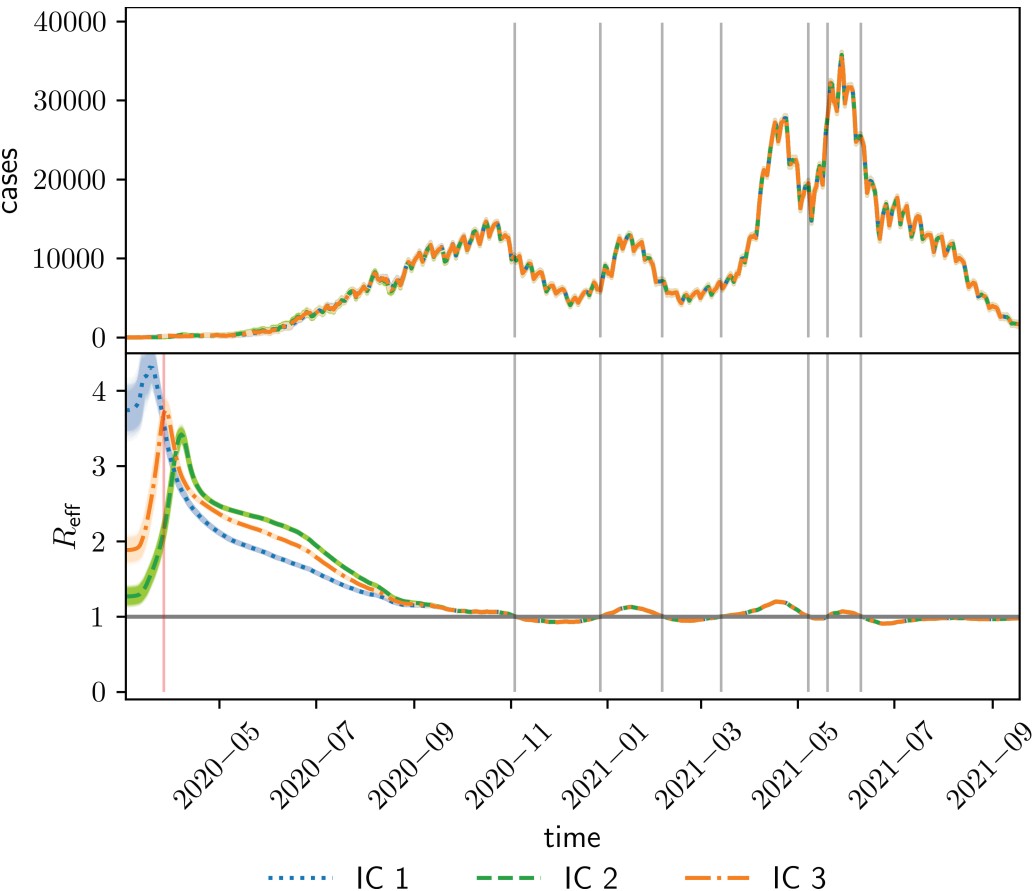

**Fig 9. Daily cases (upper panel) estimated with different initial conditions and estimated reproduction number (lower panel) using COVID-19 data of Argentina.** Shades around colored curves represent the corresponding variable spread. Vertical grey lines point out periods where $R_{\text{eff}} > 1$, and the vertical red line points out when the lockdown started.

observations are conducted, individual constant (c), linear (l) and quadratic (q) fits are performed on the last 15 days of the estimated parameterized transmission matrix values to obtain the parameter tendencies. Then, these transmission matrix tendencies are projected 30 days forward, starting from the last value of the analysis state (current day). The extrapolation of the regressed parameters is asssumed to have a standard deviation of $\sigma = 0.01$ d$^{-1}$ $t$, where $d$ is the lead time so that the parameter uncertainty increases over the lead time. Finally, 30-day forecasts are conducted with the free evolution of the model using the projected parameterized transmission matrix and starting from the current (last) analysis state. We compare the forecasts to the analysis daily cases using the entire set of observations over time as the reference. Fig 11 shows some selected forecasts up to 20 day lead time which are started over different dates (every 30 days) of the pandemic and evolved using the linear regression extrapolation of the estimated parameterized transmission matrix. Some forecasts are accurate but some diverge from what actually happened during the tendency changes of the pandemic. The orange shading indicates forecast uncertainty as given by the forecast ensemble members.

To assess the accuracy of the different forecasts, a total of 400 forecasts are conducted each started everyday and for a maximum lead time of 30 days. The forecasts cover the time window from June 2020 to the end of August 2021, featuring two peaks of the pandemic so

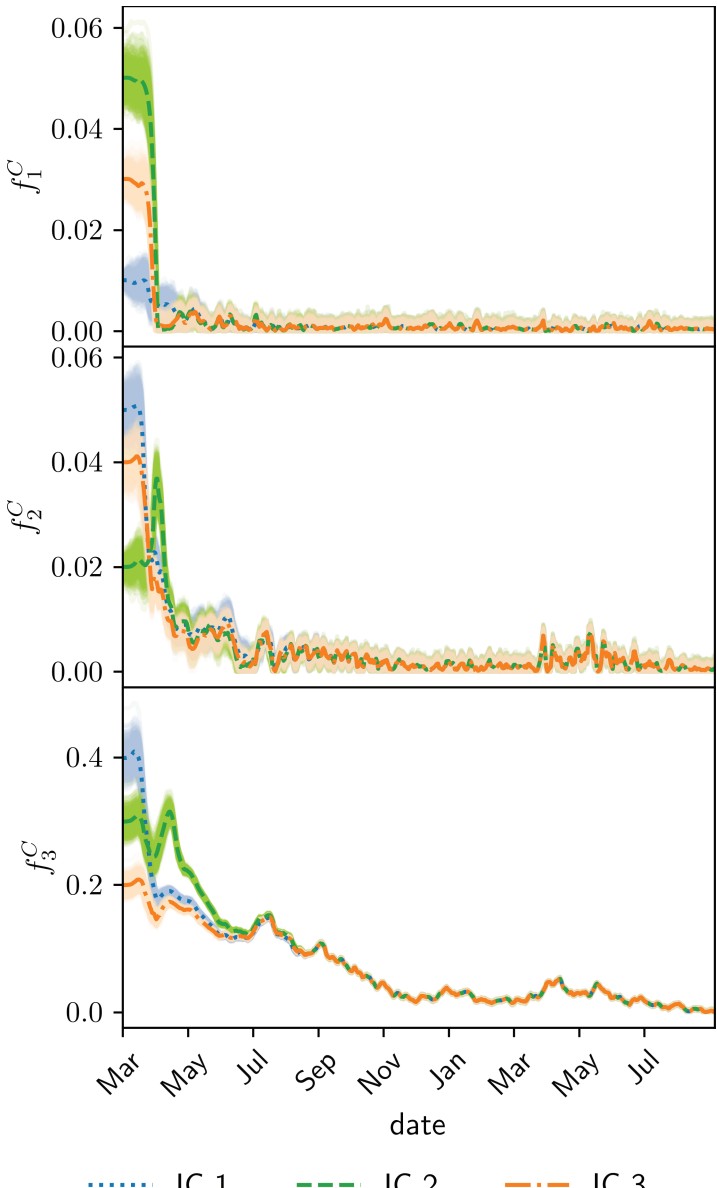

**Fig 10. Estimated fraction of deaths for young (upper), adults (medium) and senior (lower panel) age groups for different initial conditions (color lines IC1, IC2, IC3), and shades around colored curves represent the corresponding variable spread.**

that there is a wide variety of pandemic behaviors. From June 2020, a large amount of cases is detected every day so that we can safely assume a mean-field dynamics in the data assimilation framework. To examine the impact of considering the interactions among different age groups on the forecasts, we repeat the forecast experiments using a well-mixed SEIR model without age-group divisions. This model is obtained setting $n = 1$ in Eq (1). The initial condition of the well-mixed forecast is the sum along the age groups of the meta-population model.

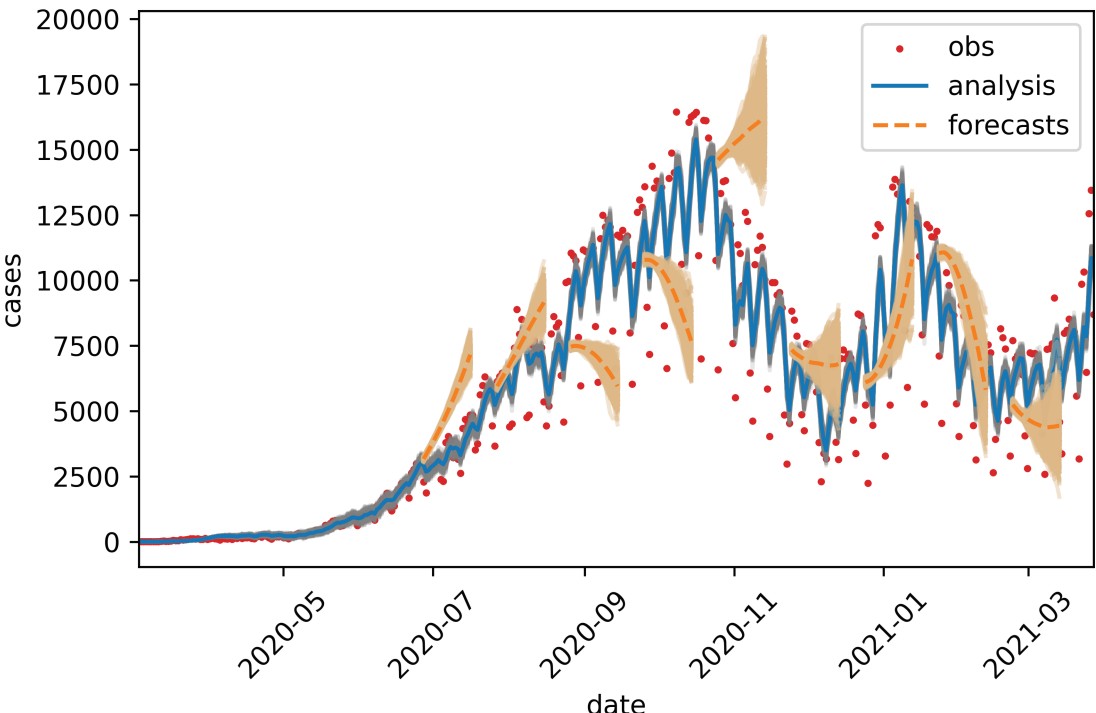

**Fig 11. Selected 20-day forecasts (orange curves) conducted at different stages of the pandemic in Argentina using the linear regression.** Initial conditions are separated over 30 days (to avoid overcluttering). Observed cumulative cases are showed as daily new cases as red dots, and analysis of cases in blue using the EnKF with multiplicative inflation. Shades around curves represent ensemble members.

Fig 12 shows the relative RMSE as a function of the lead time. The behavior of the forecasts is similar along the age groups. There is a clear advantage of the meta-population model forecasts over the well-mixed ones in all age groups. In the meta-population model, the forecast using the extrapolation of the linear regression for the parameters is the most accurate, but in the well-mixed model the constant-fit forecast outperforms the others. In both cases, the quadratic-fit forecast is less accurate.

## 5 Conclusions

In this work, we used the ensemble Kalman filter applied to a meta-population compartmental model for monitoring epidemiological parameters of the SARS-CoV-2 virus and for conducting predictions. We sequentially calibrated the model parameters using a state augmentation approach. Crucially, in contrast to recent works that use a constant transmission matrix, we provided a time-dependent parameterization of an age-dependent transmission matrix that was identifiable by the assimilation system when observations of detected cases and deaths disaggregated by age are available. This approach allows for the detection of non-trivial parameter variations and interactions between age groups which would otherwise not be captured. Additionally, the approach recovers other important epidemiological parameters such as the mortality, fraction of undocumented cases, and the effective reproduction number, the last one diagnosed using the next generation matrix. The assimilation technique serves as a valuable tool for monitoring and predicting current and future contagious diseases.

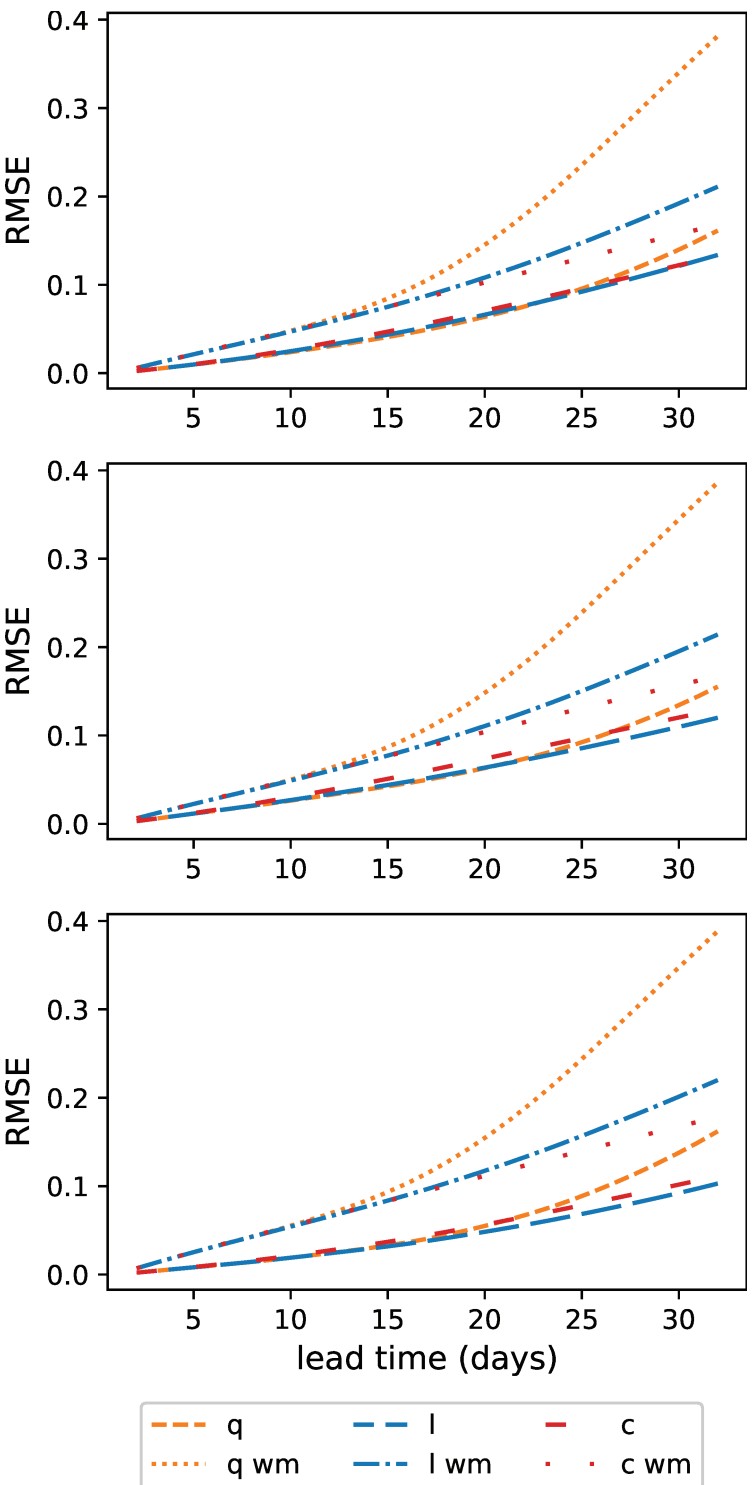

**Fig 12. Relative average root mean squared error of the quadratic (orange), linear (blue), and constant (red) forecasts for the young (upper panel), adult (middle panel) and senior people (lower panel) using the parameterized transmission matrix and the well-mixed model (labeled 'wm').** The different color lines corresponds to the quadratic, linear and constant transmission matrix tendencies (denoted q, l, c in the labels, respectively).

An assessment of the proposed technique was conducted using synthetic and real data from Argentina.

While three age groups were considered, the technique is readily adaptable to more age groups containing narrower age ranges for a more precise analysis. Attempting to estimate the full transmission matrix with only observing cases and deaths disaggregated by age groups results in the non-identifiability of matrix elements. To address this issue, we introduced a parameterization of the transmission matrix (16). This parameterization consist of a single inter-group transmission parameter, $\alpha$, which can be estimated by conducting forecasts in a validation dataset (past evolution of the pandemic up to the 'current' pandemic day) and minimizing the relative root mean squared error as a function of $\alpha$ at an a priori defined forecast lead time.

The use of the meta-population model disaggregated by age groups led to a significant improvement of the daily cases forecast accuracy for up to 30 days lead times, when compared to well-mixed model that neglect the interaction of compartments among different age groups. This highlights the critical importance of disaggregating epidemiological information in both data and models. The age-dependent forecasts hold particular relevance for the deployment of epidemiological modeling for pandemic decision-making by governments.

We assessed three parameter regression functions for the transmission matrix values. These regressed functions are then extrapolated temporally to conduct the forecasts. Up to 15-day lead times, there is practically no difference in the forecast accuracy between the three regressed functions (constant, linear and quadratic), but for longer lead times, the linear and constant regression functions results in the most accurate forecasts.

The technique could be readily implemented in different cities or states providing the population age groups and epidemic observations disaggregated by age. The framework could undergo significant improvement by including hospitalizations as an observed variable. If reliable data on check-in and check-out hospitalizations were available (which was not the case in Argentina), relevant quantities could be estimated such as average hospitalization times and use of hospital beds, as well as parameters like the fraction of hospitalizations and the fraction of intensive care.

Within the EnKF framework, we assume that errors follow a Gaussian distribution, which may not be suitable for certain model parameters. Because of this, some model parameters need to be constrained to remain within their physically meaningful range. Specifically, parameters such as the parameterized transmission matrix, the fraction of detected cases, and the fraction of deaths, are enforced to be non-negative to prevent a non-physical evolution of the model. However, this in turn conflicts with the Gaussian assumption, particularly when the spread of the parameter is close to the boundaries of their meaningful range. This is the case for the fraction of deaths in the young age group. To overcome this limitation and account for a non-Gaussian density of the near-zero parameters, a non-parametric data assimilation framework, such as the mapping particle filter [38], can be applied. Additionally, the variables are assumed to evolve smoothly, a condition that is met when dealing with a relatively large number of individuals (country-level observations). In the case of city-level populations, the behavior of the age-meta-population model within the EnKF framework may not be robust. To enhance granularity in age groups and contacts, epidemiological agent-based models can be employed. Recent works by Cocucci et al. (2022) [39] used an EnKF combined with an ABM using mean field data to infer the COVID-19 pandemic in the city of Buenos Aires, Argentina. Schneider et al. (2022) [40] employed a complex agent-based network model to assimilate synthetic data at individual level.

## Author contributions

**Conceptualization:** Santiago Rosa, Manuel A. Pulido, Juan J. Ruiz, Tadeo J. Cocucci.

**Data curation:** Santiago Rosa.

**Formal analysis:** Santiago Rosa, Manuel A. Pulido, Juan J. Ruiz, Tadeo J. Cocucci.

**Funding acquisition:** Manuel A. Pulido, Juan J. Ruiz.

**Investigation:** Santiago Rosa, Manuel A. Pulido, Juan J. Ruiz.

**Methodology:** Santiago Rosa, Manuel A. Pulido, Juan J. Ruiz, Tadeo J. Cocucci.

**Project administration:** Manuel A. Pulido.

**Resources:** Manuel A. Pulido, Juan J. Ruiz.

**Software:** Santiago Rosa, Manuel A. Pulido, Juan J. Ruiz.

**Supervision:** Manuel A. Pulido, Juan J. Ruiz, Tadeo J. Cocucci.

**Validation:** Santiago Rosa, Manuel A. Pulido, Juan J. Ruiz.

**Visualization:** Santiago Rosa.

**Writing – original draft:** Santiago Rosa, Manuel A. Pulido.

**Writing – review & editing:** Santiago Rosa, Manuel A. Pulido, Juan J. Ruiz, Tadeo J. Cocucci.

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
