## [Decision Letter · Decision Letter 0]

25 Oct 2023

PONE-D-23-20327

Transmission matrix parameter estimation of COVID-19 evolution with age compartments using ensemble-based data assimilation

PLOS ONE

Dear Dr. Rosa,

Thank you for submitting your manuscript to PLOS ONE. After careful consideration, we have decided that your manuscript does not meet our criteria for publication and must therefore be rejected.

I am sorry that we cannot be more positive on this occasion, but hope that you appreciate the reasons for this decision.

Kind regards,

Nenad Filipovic

Academic Editor

PLOS ONE

Additional Editor Comments:

Difficult to follow.

No innovation.

English should be improved.

Reviewers' comments:

Reviewer's Responses to Questions

**Comments to the Author**

1. Is the manuscript technically sound, and do the data support the conclusions?

Reviewer #1: Yes

2. Has the statistical analysis been performed appropriately and rigorously? 

Reviewer #1: I Don't Know

3. Have the authors made all data underlying the findings in their manuscript fully available?

Reviewer #1: Yes

4. Is the manuscript presented in an intelligible fashion and written in standard English?

Reviewer #1: Yes

5. Review Comments to the Author

Reviewer #1: Please see the attached file.

6. PLOS authors have the option to publish the peer review history of their article (what does this mean?). If published, this will include your full peer review and any attached files.

Reviewer #1: No

- - - - -

---

## [Author Response · Author response to Decision Letter 1]

10 Apr 2024

We would like to thank the editor and reviewer for the careful read of the manuscript. We believe the reviewer's

constructive comments and precise suggestions helped us to improve substantially this manuscript.

We state the reviewer comments and then answers to his/her comments in the file 'responses.pdf' attached to this submit.

To adress the issues raised by the editor, we improved significantly our English and pointed out the novelty of our work.

---

## [Decision Letter · Decision Letter 1]

30 Oct 2024

PONE-D-23-20327R1Transmission matrix parameter estimation of COVID-19 evolution with age compartments using ensemble-based data assimilationPLOS ONE

Dear Dr. Rosa,

Thank you for submitting your manuscript to PLOS ONE. After careful consideration, we feel that it has merit but does not fully meet PLOS ONE’s publication criteria as it currently stands. Therefore, we invite you to submit a revised version of the manuscript that addresses the points raised during the review process.

Need minor revision. Please find the reviewer comments from the mail. 

We look forward to receiving your revised manuscript.

Kind regards,

Kavikumar Jacob, Ph.D

Academic Editor

PLOS ONE

Journal Requirements:

2.Please note that PLOS ONE has specific guidelines on code sharing for submissions in which author-generated code underpins the findings in the manuscript. In these cases, all author-generated code must be made available without restrictions upon publication of the work. Please review our guidelines at https://journals.plos.org/plosone/s/materials-and-software-sharing#loc-sharing-code and ensure that your code is shared in a way that follows best practice and facilitates reproducibility and reuse.

"This work would not have been possible without the financial support of the National Agency of scientific and technology promotion (ANPCyT) of the Ministry of Science, Technology and Innovation (MinCyT) of Argentina through the projects PICT2021-I130, CORR 01 COVID FEDERAL EX-2020-38902538 and PICT2020-SERIEA-I-A."

"This study received funding from the Ministry of Science, Technology and Innovation of Argentina (PICT2021-I130 MP, JR, CORR 01 COVID FEDERAL EX-2020-38902538, SR, MP, JR, PICT2020-SERIEA-I-A JR). The sole responsibility for the content of this publication lies with the authors. The funders had no role in study design and analysis, decision to publish, or preparation of the manuscript."

5. Please update your submission to use the PLOS LaTeX template. The template and more information on our requirements for LaTeX submissions can be found at http://journals.plos.org/plosone/s/latex.

Additional Editor Comments (if provided):

Need minor revision. Please find the reviewer comments from the mail.

Reviewers' comments:

Reviewer's Responses to Questions

**Comments to the Author**

1. If the authors have adequately addressed your comments raised in a previous round of review and you feel that this manuscript is now acceptable for publication, you may indicate that here to bypass the “Comments to the Author” section, enter your conflict of interest statement in the “Confidential to Editor” section, and submit your "Accept" recommendation.

Reviewer #1: (No Response)

2. Is the manuscript technically sound, and do the data support the conclusions?

Reviewer #1: Partly

3. Has the statistical analysis been performed appropriately and rigorously? 

Reviewer #1: Yes

4. Have the authors made all data underlying the findings in their manuscript fully available?

Reviewer #1: Yes

5. Is the manuscript presented in an intelligible fashion and written in standard English?

Reviewer #1: Yes

6. Review Comments to the Author

Reviewer #1: Dear editor,

I thank the authors for addressing the issues raised in my first review report.

They made extensive revisions to the manuscript, considerably improving

its readability. However, it still needs some minor revisions before it

is considered publishable in the PLOS1.

The following issues appeared or were only noticed in the revised version of the manuscript.

MINOR CONCERNS

1) The authors should cite and/or include in the text the equations of motion of

the well-mixed SEIR model mentioned in line 542. This could be in the

supplementary material. I believe the parameters could be obtained from Model

(1) and Table 1. The parameters of this simplified model should be informed as

well. Are the parameters also time dependent? I think the same type of

algorithm is used in both systems (with the meta-population model and the

well-mixed SEIR model).

2) The time-step evolution operator of the dynamical model M can be obtained

clearly from equation (1).

It is still not very clear which parameters constitute the parameter vector

theta_k in the stochastic eq. (12). From the figures, it seems the infection

coefficients, the gammas, and the fractions f^C_j, with j=1,2,3, jointly

constitute the vector theta_k. What the vector theta_k is exactly should be

stated in the manuscript.

3) How is the observational covariance matrix R just below eq.(4) obtained?

4) To make the manuscript more readable, just below eq. (7),

I suggest explaining what the forecast step means, perhaps referencing Algorithm 1.

5) I think the authors should include the step in which y_k is obtained

in the Algorithm 1 description.

6) The authors could refer to https://en.wikipedia.org/wiki/Detailed_balance

for more information on the detailed balance importance to Statistical Physics.

7) The authors could inform the reader what is the specific matrix H used just below equation (11).

8) When printed in black and white it is hard to see the differences of the

darker solid/dashed lines. This is seen most notably in figs. 2, 4, 8, 10, 12.

I recommend the authors use different thicknesses or line styles in addition to

color so as to better distinguish them.

9) In the end of the Introduction section, I think the authors inadvertently

erased the paragraph with the short description of the remaining sections of the

manuscript.

10) I did not find the added reference comparing MCMC models and data

assimilation inference in line 489.

TYPOS

In line 71, 'accurate' instead of 'accurately'

In line 101, 'This model classifies' instead of 'This model classify'

In line 213, 'Observation' instead of 'Obervation'.

In line 500, perhaps the authors meant 'inertia' instead of 'innerce'.

In line 585, 'linear, and quadratic' instead of 'linear quadratic'

CONCLUSIONS

I believe the manuscript will have a stronger impact on the epidemics modeling

community now. The borrowed techniques from weather forecasting may prove

invaluable tracking future epidemics.

7. PLOS authors have the option to publish the peer review history of their article (what does this mean?). If published, this will include your full peer review and any attached files.

Reviewer #1: No

---

## [Author Response · Author response to Decision Letter 2]

16 Dec 2024

Dear Dr. Jacob and anonymous reviewer,

Thank you for taking the time to review our manuscript. We have addressed the journal's requirements as follows:

1) We carefully revised the manuscript to ensure full compliance with the style guidelines outlined in the formatting documents.

2) The repository containing all relevant codes used in our experiments has been made public. Details are provided in the section Code and Data Availability.

3) The Acknowledgments section has been removed as requested. The funding information remains unchanged.

4) All raw and curated data have been made publicly available, as stated in the section Code and Data Availability.

5) We used the official PLOS template to generate the PDF files.

As for the reviewer’s concerns, we have addressed all the points and implemented your suggestions. The details are outlined in the attached response to the reviewer, as they are too extensive to include here.

If there are any additional issues or specific concerns with our manuscript, please let us know.

Kind regards,

Santiago Rosa

---

## [Decision Letter · Decision Letter 2]

16 Jan 2025

Transmission matrix parameter estimation of COVID-19 evolution with age compartments using ensemble-based data assimilation

PONE-D-23-20327R2

Dear Dr. Rosa,

We’re pleased to inform you that your manuscript has been judged scientifically suitable for publication and will be formally accepted for publication once it meets all outstanding technical requirements.

Kind regards,

Matthew Chin Heng Chua

Academic Editor

PLOS ONE

Additional Editor Comments (optional):

the authors have undergone multiple rounds of revision and have addressed the reviewers' comments satisfactory. it is in my opinion that the manuscript is of sufficient standard for acceptance.

Reviewers' comments:

Reviewer's Responses to Questions

**Comments to the Author**

1. If the authors have adequately addressed your comments raised in a previous round of review and you feel that this manuscript is now acceptable for publication, you may indicate that here to bypass the “Comments to the Author” section, enter your conflict of interest statement in the “Confidential to Editor” section, and submit your "Accept" recommendation.

Reviewer #1: All comments have been addressed

2. Is the manuscript technically sound, and do the data support the conclusions?

Reviewer #1: Yes

3. Has the statistical analysis been performed appropriately and rigorously? 

Reviewer #1: I Don't Know

4. Have the authors made all data underlying the findings in their manuscript fully available?

Reviewer #1: Yes

5. Is the manuscript presented in an intelligible fashion and written in standard English?

Reviewer #1: Yes

6. Review Comments to the Author

Reviewer #1: I am fully satisfied with the authors replies.

I think they addressed all the issues I raised

in my second review report.

7. PLOS authors have the option to publish the peer review history of their article (what does this mean?). If published, this will include your full peer review and any attached files.

Reviewer #1: No

---

## [Editor Report · Acceptance letter]

PONE-D-23-20327R2

PLOS ONE

Dear Dr. Rosa,

I'm pleased to inform you that your manuscript has been deemed suitable for publication in PLOS ONE. Congratulations! Your manuscript is now being handed over to our production team.

Kind regards,

on behalf of

Prof. Matthew Chin Heng Chua

Academic Editor

PLOS ONE